# 3D hydrogeological parametrization using sparse piezometric data

Dimitri Rambourg[1], Raphaël Di Chiara[1], Philippe Ackerer[1]

[1]Institut Terre et Environnement de Strasbourg, Université de Strasbourg/EOST/ENGEES, CNRS UMR 7063, 5 rue Descartes, Strasbourg F-67084, France

5    *Correspondence to*: Dimitri Rambourg (d.rambourg@unistra.fr)

**Abstract.** When modelling contamination transport in the subsurface and aquifers, it is crucial to assess the heterogeneities of the porous medium, including the vertical distribution of the aquifer parameter. This issue is generally addressed thanks to geophysical investigations.

As an alternative, a method is proposed using estimated hydraulic parameters from a 2D calibrated flow model (solely reliant 10    on piezometric series) as parameterization constraints for a 3D hydrogeological model. The methodology is tested via a synthetic model, ensuring full knowledge and control of its structure. The synthetic aquifer is composed of five lithofacies, distributed according to a sedimentary pattern, and functions in an unconfined regime. The level of heterogeneity for hydraulic conductivity spans three orders of magnitude. It provides the piezometric chronicles used to inverse 2D flow parameter fields and the lithological logs used to interpolate the 3D lithological model. Finally, the parameters of each facies (hydraulic 15    conductivity and porosity) are obtained through an optimization loop, that minimizes the difference between the 2D calibrated transmissivity and the transmissivity computed with the estimated 3D facies parameters.

The method estimates values close to the known parameters, even with sparse piezometric and lithological data sampling. The maximal discrepancy is 45 % of the known value for the hydraulic conductivity and 6 % for the porosity (mean error 26 % and 3 %, respectively). Although the methodology does not prevent from interpolation errors, it succeeds in reconstructing 20    flow and transport dynamics close to the control data. Due to the inherent limitations of the 2D inversion approach, the method only applies to the saturated zone at this point.

## 1 Introduction

To simulate contamination transport in the subsurface and aquifers, it is crucial to assess and reliably describe the heterogeneities of the porous medium. The development of inverse methods in recent decades is mainly based on two-25    dimensional flow models and focused on the horizontal structure of heterogeneities with the collection of piezometric data as a cornerstone (Poeter & Hill, 1997; Carrera et al., 2005; Hendricks Franssen et al., 2009). But the latter is less sensitive to the vertical structure of the aquifer, leaving its estimation dependent on complex and expensive field methods – e.g. pumping tests (De Caro et al., 2020), tracer tests (Linde et al., 2006), electrical resistivity (Coscia et al., 2011; Priyanka & Mohan Kumar, 2019), radar tomography (Boni et al., 2020), self-potential methods (Eppelbaum, 2021), crosshole testing (Klotzsche et al.,

2013; Doetsch et al., 2010), hydraulic tomography (Sanchez-León et al., 2015; Luo et al., 2020; Fischer et al., 2020) – and/or laboratory analysis – e.g. grain-size analysis from core samples (Marini et al., 2018) and ex-situ permeability tests (Zhang & Brusseau, 2005). The collection of these information, describing the vertical heterogeneity of the aquifer, allows the development of 3D inversion techniques. For example, some successful methods combine direct parameter quantification and stochastic geological modelling (Guadagnini et al., 2004; Fu & Gómez-Hernández, 2008; Cardiff & Kitanidis, 2009), others

incorporate water head data and more advanced geophysical measurements to the (joint) inversion procedure (Straface et al., 2011; Lee & Kitanidis, 2014).

Hydrogeological models are usually two-dimensional and transmissivities are estimated through model calibration. 2D models are easier to handle considering field work, parameter measurements, data manipulation and calibration than 3D models. We question here the possibility of transferring data from a 2D calibrated hydrogeological model to a 3D configuration. Viaroli et

al. (2019) recently employed a simplified 2D model to specify the boundary conditions and recharge of a 3D model already designed by other means. Incidentally, the design and parameterization of the 3D model itself is even more seldom independent of geophysical methods. In this line, we propose an original method using data from a 2D calibrated flow model (solely reliant on piezometric time series) as parameterization constraints for a 3D hydrogeological model resulting from interpolation of borehole data. The use of 2D calibrated transmissivities allows our technique to be completely unrelated to geophysical

methods, and less heavy-computational than 3D joint inversion approaches. Moreover, the articulation of the method also allows to take advantage of a pre-existing 2D calibrated model, if any.

The method is tested on a synthetic test case constituted by five hydrofacies, distributed according to a sedimentary pattern, with a level of heterogeneity for hydraulic conductivity spanning three orders of magnitude. This work can be considered as an improvement of the method proposed by Harp et al. (2008), who also tested the combination of 2D inversion and an

interpolation method, but on a two-dimensional transect model composed of only two facies.

In order to evaluate our methodology's robustness, it is first carried out with a very profuse data sampling (piezometric for the 2D inversion; lithological for the 3D model interpolation), assessing the consistency between the different numerical codes. Second, a sparser sampling is tested to approximate more realistic field conditions.

The detail of the methodology is described in Sect. 2, including the synthetic data framework, the mathematical background

of the tools used, and the link between them. The results for both samplings concerning the inversions, the facies interpolation, and the final model outputs (in terms of parameter, piezometric series, and contamination plumes) are discussed in Sect. 3.

## 2 Materiel and methods

The methodology we propose and analyse in this paper is the following:

1. Estimates of transmissivity and porosity from a 2D calibrated flow model based on piezometric heads. These transmissivities exist for each element/cell of the 2D mesh. It is a huge data set constrained by the piezometric heads.

2. Analysis of the aquifer lithology at boreholes. Lithology is usually described at each borehole. It provides a qualitative description of aquifer heterogeneity. This qualitative description can be interpreted in terms of facies. This description is used here to define an optimal number of facies that have been identified within the aquifer.

3. The 3D discretization of the aquifer is a vertical extension of the 2D model which avoids interpolation of the 2D parameters. Facies are interpolated over the 3D domain based on the borehole local data.

4. The hydraulic conductivity and porosity for each facies are estimated through optimization using the 2D data, which are considered as vertical integrations of the 3D data. Optimization is required because the number of unknowns is quite small (twice the number of facies) compared to the number of constraints (twice the number of elements of the 2D flow model at the most). Of course, the constraints are correlated through the flow model and cannot be considered independent.

To evaluate this approach, we built a synthetic test case (Fig. 1) generated by:

1. 3D aquifer design.

2. Computation of the 3D flow using the software TRACES (Hoteit & Ackerer, 2004).

3. Selection of representative head data considered constraints for the 2D inversion.

4. Estimation of transmissivities and mean vertical porosity by a 2D flow model calibration based on the selected head data using PINOGRI (Rambourg et al., 2020).

5. Selection of representative boreholes for lithological data and facies definition.

6. Design of the 3D facies distribution using a geostatistical interpolator (GemPy - de la Varga et al., 2019) or a deterministic interpolator (Splines - Lee et al., 1997).

7. Estimation of each facies hydrodynamic parameters (hydraulic conductivity, porosity) using an optimization procedure constrained by the 2D calibrated values.

8. Comparison of local hydrodynamic parameters, simulated water heads, and concentrations between the "true" aquifer and the reconstructed (estimated) aquifer.

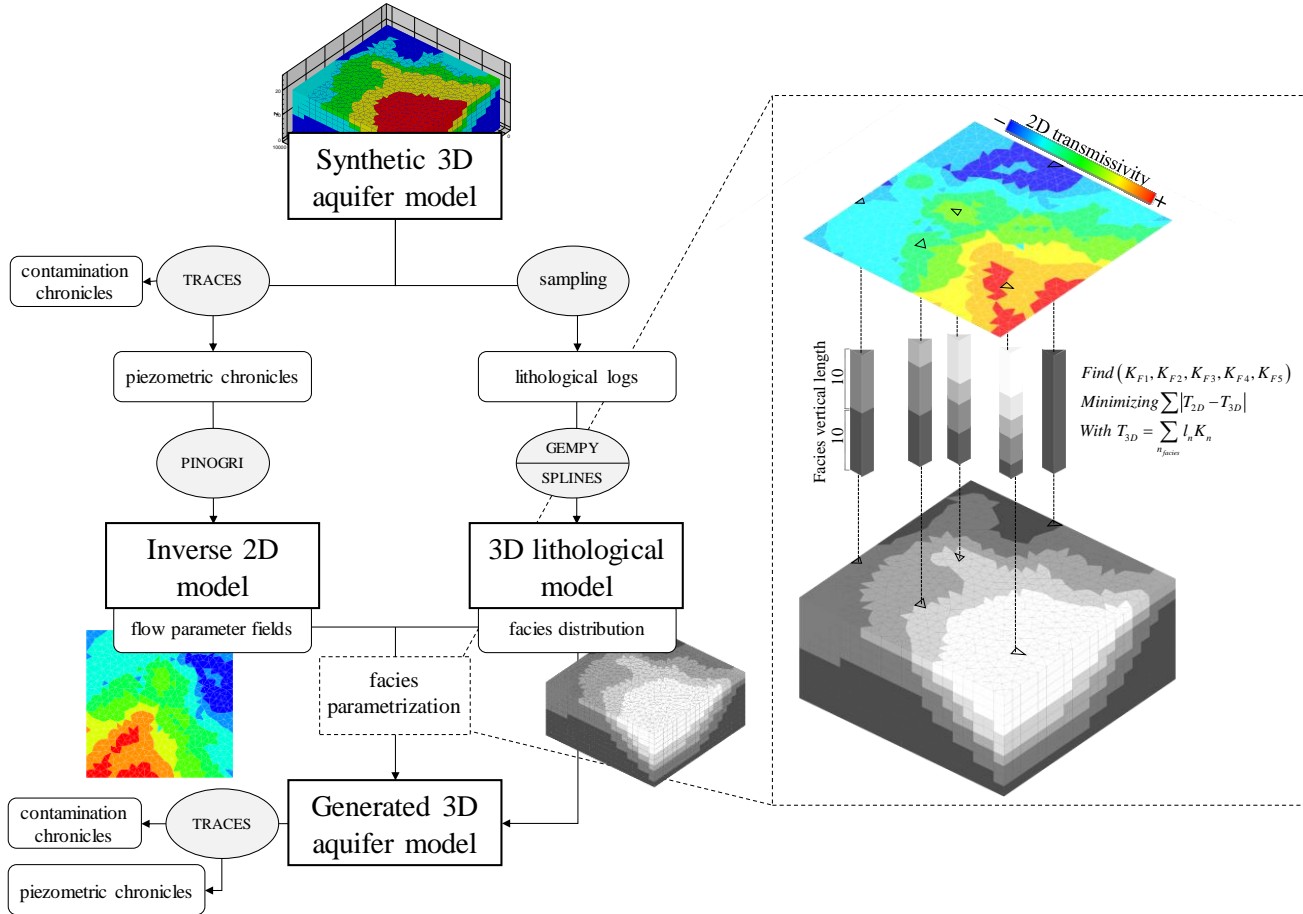

Find $\left(K_{F1}, K_{F2}, K_{F3}, K_{F4}, K_{F5}\right)$

Minimizing $\sum \left| T_{2D} - T_{3D} \right|$

With $T_{3D} = \sum_{n_{facies}} l_n K_n$

85

**Figure 1: Methodology flowchart – TRACES: Transport of Radioactive Elements in Subsurface (Hoteit & Ackerer, 2004); PINOGRI: Parameter Inversion Numerically Optimized for Groundwater Issues (Rambourg et al., 2020); GEMPY: Open-source 3D geological modelling (de la Varga et al., 2019); SPLINES: QGIS/SAGA multilevel b spline interpolation (Lee et al., 1997).**

The computations are run on a PC with Intel(R) Core(TM) i7-6700 CPU @ 3.40 GHz processor and 16 Go RAM.

**2.1 Synthetic three-dimensional dataset**

### *2.1.1 The aquifer model*

The synthetic aquifer consists of five hydrogeological facies (also referred to as hydrofacies) distributed along a sedimentary pattern over a 10x10 km area and 20 m depth (Fig. 2).

Each hydrofacies is characterized by a hydraulic conductivity five times higher than the underlying facies (Tab. 1). Their 95 porosity is less heterogeneous as it is defined in the range of permeable sedimentary materials (10 %−30 %). In practice, a hydrofacies is defined by clustering lithofacies with comparable hydrodynamic properties. The limitations and pitfalls inherent in this step are not addressed in this study, where it is assumed to be flawless.

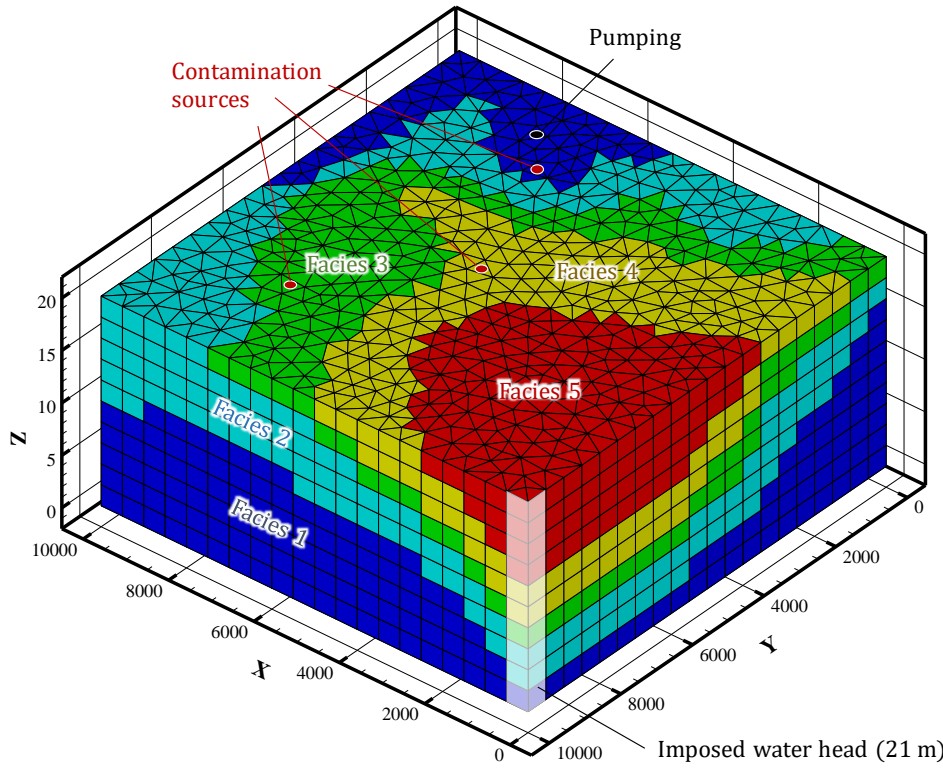

**Figure 2: Three-dimensions facies distribution of the synthetic aquifer and aquifer discretization – the black dot is the location of the pumping well, the red dots the location of the contaminant sources.**

Hydraulics boundary conditions are of null-Neumann type (no flux) except the northwest corner where a 21 m head is imposed (Dirichlet boundary), acting as the outlet of the aquifer. A constant pumping well (18 m$^3$.h$^{-1}$) is positioned in the southeast part of the model, intercepting the whole thickness. For solute transport, a zero solute flux is prescribed at the boundary, except at the aquifer outlet, where the outflow is considered as purely advective.

**Table 1: Synthetic hydrofacies parameters.**

| Facies | F1 | F2 | F3 | F4 | F5 |
|---|---|---|---|---|---|
| Hydraulic conductivity [m.s⁻¹] | $1 \times 10^{-5}$ | $5 \times 10^{-5}$ | $2.5 \times 10^{-4}$ | $1.25 \times 10^{-3}$ | $6.25 \times 10^{-3}$ |
| Effective (kinematic) porosity | 10 % | 15 % | 20 % | 25 % | 30 % |
| Global proportion[1] | 51 % | 26 % | 10 % | 7 % | 6 % |
| Superficial proportion[2] | 11 % | 19 % | 23 % | 24 % | 23 % |
| Recharge [mm.y⁻¹] (% of rainfall) | 21 (3 %) | 43 (6 %) | 64 (9 %) | 86 (12 %) | 107 (15 %) |

[1] Proportion of facies at the scale of the whole domain        [2] Proportion of facies within the surface elements

The aquifer is exclusively fed by rainfall (720 mm.y$^{-1}$ on average), which is assumed homogeneous over the whole area. However, five different recharge patterns are imposed according to the most superficial facies, whose hydrodynamic parameters greatly influence the amount and dynamics of water infiltration. Thus, recharge zone 5 (formed by the most permeable surface facies) is subject to major and fast infiltration, in contrast to zone 1 (least permeable), where the seepage signal is very attenuated and spread over time (see Sect. 3).

### 2.1.2 Piezometric and contamination reference data generation

The behaviour of groundwater and dissolved contamination is computed using TRACES (Transport of Radioactive Elements in Subsurface) software (Hoteit & Ackerer, 2004), a numerical code written in FORTRAN 90 for the simulation of flow and reactive transport in saturated/unsaturated porous media.

The three-dimensional flow model is the combination of the conservation of mass and Darcy's laws, generalized to also apply to the unsaturated zone (Darcy-Buckingham law), resulting in the Jacob-Richards equation (Eq. 1):

$$\frac{\partial \theta}{\partial t} + s \frac{\theta}{\phi} \frac{\partial h}{\partial t} - \nabla \cdot (\mathbf{K} \nabla h) = f \tag{1}$$

where $\theta$ and $\phi$ are the water content [-] and porosity [-], respectively, necessary to deal with the unsaturated zone. $s$ and $\mathbf{K}$ are the specific storage coefficient [m$^{-1}$] and hydraulic conductivity tensor [m.s$^{-1}$], respectively. $h$ is the water head [m], and $f$ is the sink-source term [s$^{-1}$].

To limit inconsistencies with the 2D inversion (where unsaturated flow is not addressed via a physical model), the 3D model is reduced to a fully saturated approach. Therefore, flow equation (Eq. 1) is simplified and shown with the adequate initial and boundary conditions as Eq. 2.

$$\begin{cases} \mathbf{S} \frac{\partial h}{\partial t} - \nabla \cdot (\mathbf{T} \nabla h) = F \\ h(\mathbf{x}, 0) = h_0(\mathbf{x}) & \mathbf{x} \in \Omega \\ h(\mathbf{x}, t) = h_D(\mathbf{x}, t) & \mathbf{x} \in \Gamma_D \quad t \in [0, T] \\ \mathbf{T} \nabla h(\mathbf{x}, t) \cdot \mathbf{n} = q_N(\mathbf{x}, t) & \mathbf{x} \in \Gamma_N \quad t \in [0, T] \end{cases} \tag{2}$$

where $\mathbf{S}$ is the storativity [-], equivalent to the effective porosity in an unconfined context. $\mathbf{T}$ is the transmissivity [m$^2$.s$^{-1}$], the integration of the hydraulic conductivity over the vertical of the model. $F$ [m.s$^{-1}$] is the sink-source term, $\mathbf{x}$ is a position in $\Omega$, the model domain, and $h_0(\mathbf{x})$ represents the initial conditions. $\Gamma_D$ and $\Gamma_N$ are partitions of the domain boundaries that correspond to Dirichlet and Neumann conditions, respectively, and $\mathbf{n}$ is the unit vector normal to the boundary, counted positive outward. $h_D(\mathbf{x}, t)$ is the prescribed head value at the Dirichlet boundaries, and $q_N(\mathbf{x}, t)$ is the prescribed flux at the Neumann boundaries, both defined at each time $t$ of the simulated period $T$.

The assumption of a locally constant transmissivity is satisfied, with water head variations of a maximum of 5.2 % (and 3.5 % on average) of the local mean water head.

TRACES addresses the migration of contaminants via an advection-dispersion/diffusion equation, supporting adsorption, precipitation, and degradation (reactive transport) phenomena. However, the study considers one inert species, giving form to Eq. 3.

$$\begin{cases} \frac{\partial(\theta C)}{\partial t} - \nabla \cdot (\theta \mathbf{D} \nabla C + \mathbf{q}C) = Q & \mathbf{q} = -\mathbf{K}\nabla h \\ C(\mathbf{x}, 0) = C_0(\mathbf{x}) & \mathbf{x} \in \Omega \\ -(\mathbf{D}\nabla C \cdot \mathbf{n})A(t) + B(t)C = q(t) & t \in [0, T] \end{cases} \tag{3}$$

where $C$ is the solute concentration [kg.m$^{-3}$], $\mathbf{D}$ is the dispersion/diffusion tensor [m$^2$.s$^{-1}$], $\mathbf{q}$ is Darcy's velocity [m.s$^{-1}$], and $Q$ is the solute sink-source term [kg.m$^{-3}$.s$^{-1}$]. $C_0(\mathbf{x})$ is the initial concentration; $A(t)$, $B(t)$, and $q(t)$ are the parameters to define the boundary conditions (see Hoteit & Ackerer, 2004).

The dispersion and diffusion parameters are set identically for all the facies (Tab. 2).

Table 2: Transport parameters.

| Parameter | Longitudinal dispersivity [m] | Transversal dispersivity [m] (horizontal) | (vertical) | Molecular diffusion [m$^2$.s$^{-1}$] |
|---|---|---|---|---|
| Value | 1 | 0.1 | 0.1 | $10^{-9}$ |

These parameters are transferred into the dispersion/diffusion tensor as shown in Eq. 4.

$$\theta \mathbf{D} = \theta D_m \cdot \tau + D_T \|\mathbf{q}\| \delta_{ij} + \frac{(D_L - D_T)q_i q_j}{\|\mathbf{q}\|} \quad \text{with} \quad \tau = {\theta^{7/3}}/{\phi^2} \tag{4}$$

where $D_m$ is the molecular diffusion coefficient [m$^2$.s$^{-1}$], and $\tau$ is the tortuosity factor of the porous medium [-] (according to Millington & Quirk, 1961). $D_T$ and $D_L$ are the transversal and longitudinal dispersivities [m], respectively, while $\delta$ is the Kronecker function, with $i$ and $j$ the position indexes in the tensors.

Flow and transport equations implemented in the code TRACES are solved under transient or steady state computation in 2D or 3D heterogeneous domains. Mixed hybrid finite elements are used to solve the flow equation and the diffusive/dispersive component of the transport. A mass lumping formulation is used to limit the occurrence of numerical oscillations. The advective part of the transport is solved using discontinuous Galerkin finite element which also prevents numerical oscillations in the simulations and strongly limits numerical diffusion. These numerical schemes ensure an exact mass balance at the element level and are very flexible in space discretization (triangular or quadrangular element in 2D, tetrahedral, prismatic of hexahedral elements in 3D).

As shown in Fig. 2, the model consists of 10,440 triangular prisms, 6,193 nodes, and 27,544 facies. The horizontal edges have a characteristic length of 500 m, while the vertical edges are 2 m long.

The initial state of the water table is derived from a preliminary steady-state calculation involving averaged recharge. The aquifer is initially uncontaminated ($C_0 = 0$ over the whole domain) and undergoes a pollution episode from three surface sources (Fig. 2), each discharging 0.1 g.s$^{-1}$ for the first 24 hours of the simulated time.

### 2.1.3 Structural and piezometric reference data sampling strategies

Structural (hydrofacies) and piezometric data are sampled following two subsequent strategies in order to validate the method (Fig. 3).

First, the methodology is conducted with a very dense data set (400 control points) to assess its potential under ideal conditions and verify the numerical approaches' compatibility. The piezometric chronicles used for the inversion cover the nine years of simulation (see Sect. 3).

Second, a sparser dataset is extracted to evaluate the method in more realistic conditions. The control points are reduced to 40, and the piezometric chronicles are shortened randomly (down to 2 years). Meanwhile, the hydrofacies logs extracted at the location of the control points are kept in their integrity.

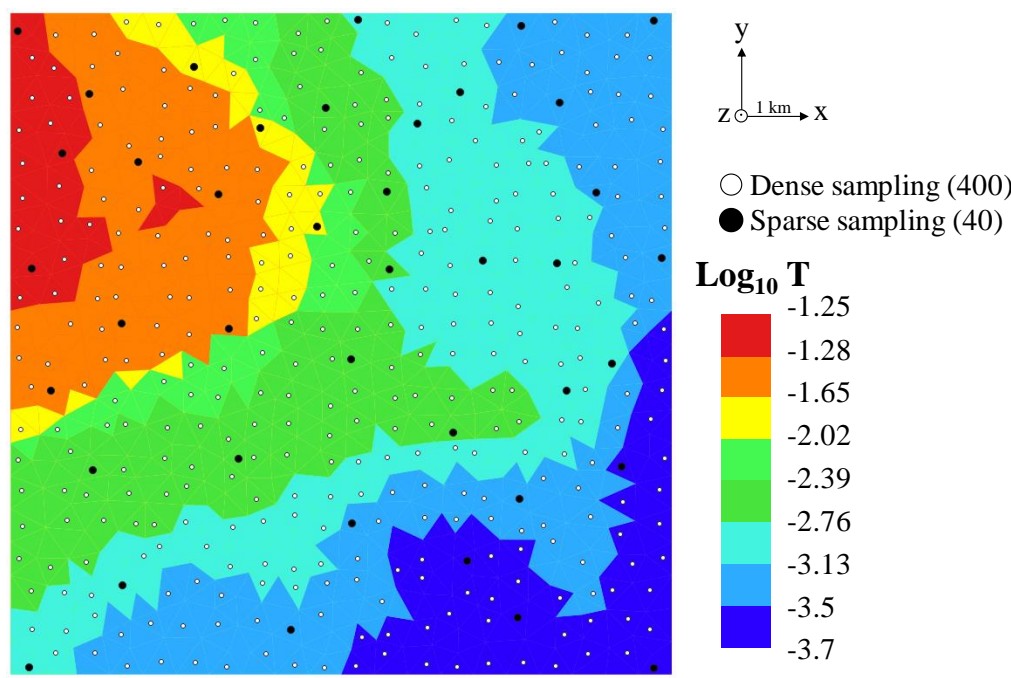

**Figure 3: Transmissivity map of the aquifer and location of the wells for sparse (black dots) and dense (white dots) samplings.**

The sparse sampling is pseudo-random so that each recharge area is covered. The points at each corner of the model are included in both sampling strategies to avoid extrapolation issues. The sparse sampling accounts for 10 % of the lithological information of the dense dataset and only 6 % of the water head data.

## 2.2 Two-dimensional flow model inversion

The sampled piezometric data are used as inversion constraints for the PINOGRI (Parameter Inversion Numerically Optimized for Groundwater Issues, see Rambourg et al., 2020) software, developed at ITES (Strasbourg). Water heads being less sensitive to the vertical heterogeneities of the porous media, the inversion approach is restricted to a two-dimensional scale, where heads are vertically averaged. This step results in the estimation of transmissivity and average porosity fields at the scale of each mesh of the model. The inversion procedure consists of minimizing an objective function (the quadratic difference between measured and computed piezometric heads) with parameter optimization guided by a gradient descent method.

Although piezometric data is subject to uncertainty in a field context, we do not address this aspect in the present study and the water heads measurements errors are considered as negligible.

### 2.2.1 The flow model

Two-dimensional groundwater flow in the aquifer is described by a diffusion-type equation, akin to the TRACES approach, but with a constant head over depth assumption (Dupuit-Forchheimer's hypothesis) reducing the problem's dimension.

The mathematical model is solved by a two-dimensional nonconforming finite element method (Crouzeix & Raviart, 1973), ensuring flux continuity, mass balance (like the finite volume method), flexibility in geometry, and rigorous computation of full tensor transmissivity (like conforming finite elements, as stated by Ackerer et al., 2014). The time discretization scheme is implicit, giving the direct problem the form of Eq. 5.

$$\mathbf{A}h^t = F^{t-1} \tag{5}$$

where $h$ is the water head vector at the calculation time $t$, and $F$ is the sink-source term vector produced at the previous time step. $\mathbf{A}$ is the flow coefficient matrix, depending on the mesh geometry and the parameter vector.

### 2.2.2 The inverse problem

The groundwater flow parameters are estimated through the minimization of an objective function (Eq. 6) based on weighted least square (Carrera & Neuman, 1986; Tarantola, 2005).

$$J(\mathbf{P}) = (h(\mathbf{P}) - h^*)^{\mathrm{T}} \mathbf{W}^{-1}(h(\mathbf{P}) - h^*) \tag{6}$$

where $J$ is the objective function, $\mathbf{P}$ represents the vector of the parameters to be estimated, $h^*$ is the measured piezometric head (obtained from vertical averaging of 3D sampled data), and $h$ is the corresponding simulated values. T is the transpose operator, and $\mathbf{W}$ is the weighting matrices, depending on measurement errors, able to prioritize optimization effort on specific locations. Therefore, in this study, all data are weighted equally. Moreover, because no a priori hydraulic parameters information is added in the study, the objective function does not include the plausibility criterion of the Maximum Likelihood approach.

Due to the great number of parameters and measurements, the minimization of the objective function is led by a quasi-Newton method which is less time-consuming compared to Gauss-Newton and other Jacobian-based approaches (Kitanidis & Lane, 1985). The gradient and an approximate Hessian of the objective function are calculated using the discrete adjoint state method (Carter et al., 1974) and the limited memory BFGS (Broyden-Fletcher-Goldfarb-Shanno) algorithm (Byrd et al., 1995), respectively. In our case, the adjoint state method is used to compute the gradient of the objective function (required by the L-BFGS algorithm) as an optimization problem of a Lagrangian, constrained by the head values obtained from the direct calculation. On another hand, instead of calculating the sensitivity coefficients for each parameter at each iteration required by Newton methods, the L-BFGS algorithm (Quasi-Newton) approaches a Hessian approximation by converging an initial matrix (e.g. the identity matrix) according to the results from a limited number of previous iterations. As the parametrization of the inversion can lead to a high number of degrees of freedom, this set of techniques has been found more efficient than standard sensitivity approaches (Townley & Wilson, 1985). Finally, three stopping criteria are set to end the algorithm: (i) the objective function J or its gradient is sufficiently low, (ii) the adjustment of the parameters **P** or the decrease of $J$ between two iterations is too small, or (iii) the number of iterations has reached a user-set maximum value.

Incidentally, the inverse problems generally suffer from being ill-posed, i.e. the number of data (locally known piezometry) is too low compared to the number of unknowns (hydraulic conductivity and porosity at the scale of each mesh element). This leads to issues of non-uniqueness and instability of solutions. One way to limit these inconveniences is to reduce the number of unknowns via a parametrization technique. In PINOGRI, the parameter spatial pattern is inferred using an Adaptive Multiscale Triangulation (AMT - Majdalani & Ackerer, 2011; Hassane & Ackerer, 2017). The parameters, borne by the vertices of the AMT mesh, are interpolated into each element of the calculus mesh (see Fig. 4). If during the inversion process, the minimization criteria are not met at the scale of a parameter cell, the latter is divided into four, increasing the optimization's degree of freedom. Refinement is halted either when the objective function at the element level drops below a user-defined threshold, when the number of iterations reaches a user-defined maximum, or when the last iteration fails to produce a better optimization than the previous one. This adaptive approach allows more flexibility and need less preconceptions about the model structure than fixed parametrizations, such as zonation or interpolations. A detailed description of the mathematical developments and the algorithm can be found in Ackerer et al. (2014) and Hassane & Ackerer (2017).

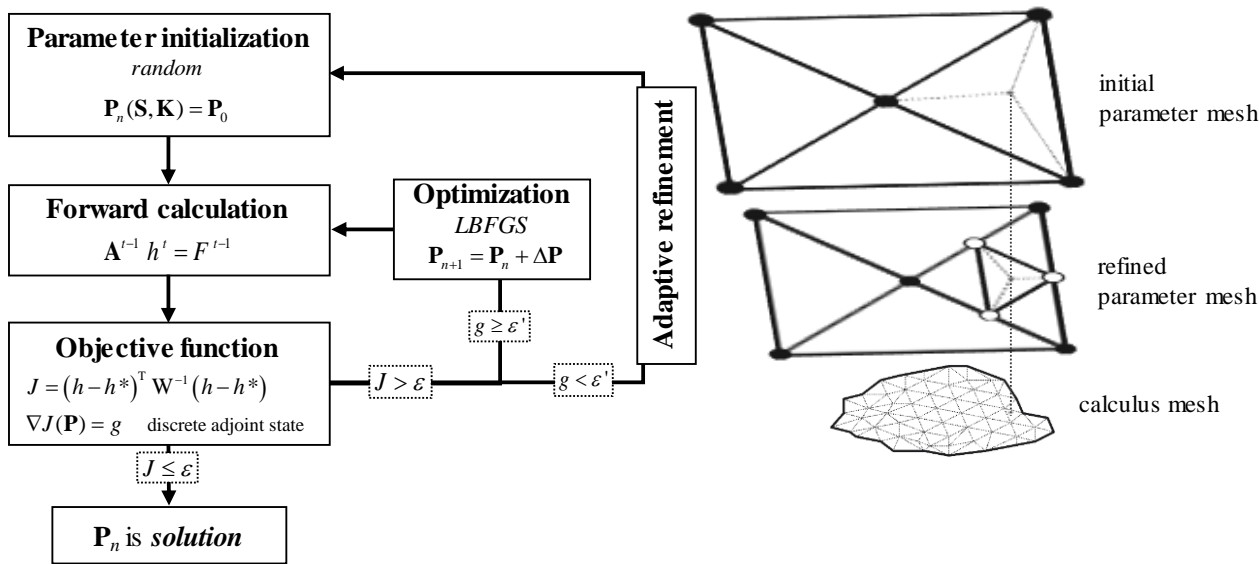

Figure 4: Inversion algorithm (adapted from Rambourg et al., 2020) – on the left, the general algorithm; on the right, the adaptive parametrization and its first refinement pattern.

A maximum of three adaptive multiscale iterations is set to ensure a satisfactory calibration while preventing overparameterization.

The boundary conditions of the 2D approach are exactly the same as for the 3D synthetic model. In contrast, the initial conditions cannot be integrally transposed, the knowledge of the water head being limited by data sampling. Thus, the initial water head for the 2D approach is derived from preliminary steady-state inversions constrained by time-averaged water head data.

The parameter bounds of the inversion are $5 \times 10^{-6} - 5 \times 10^{-2}$ for the hydraulic conductivity [m.s$^{-1}$] and 6 %–30 % for the porosity.

## 2.3 Facies interpolation

For comparison purposes, two interpolation methods are used to reconstruct the 3D facies distribution, both based on sampled lithological logs. The interpolation estimates the distribution between control points, using statistical dependency of measures in the case of geostatistical methods, or drawing geometric surfaces independently of the spatial statistical repartition in the case of deterministic methods. For both approaches, the resulting 3D models contain only qualitative (indicator) data. The interpolation methods selected are not exclusive, as the general methodology presented can accommodate any interpolation tool that produces a 3D facies model.

### 2.3.1 GemPy (geostatistical interpolation)

GemPy (de la Varga et al., 2019) is an open-source 3D geomodelling package written in Python. It specializes in the reconstruction of stratigraphic series, with the possibility of modelling complex environments by adding faults, folds, plutonic

intrusions, and other anomalies. The GemPy's mathematical background is a development of the work of Lajaunie et al. (1997;
Lajaunie et al., 1997; Calcagno et al., 2008), using universal cokriging methods to interpolate potential fields (scalar fields).

Kriging covers a set of exact (unbiased) linear estimation techniques that minimize the estimation variance using the variogram, a function representing the correlation level of a random variable as a function of distance. Initially limited to stationary variables (simple and ordinary kriging), universal kriging has extended the use of this type of geostatistical methods to non-stationary variables. Eventually, cokriging not only uses the spatial correlation of a variable with itself but also incorporates
the cross-correlations between 2 or more random variables.

In the software, the interpolation concerns two types of data: isosurfaces (including the interface between stacked lithologies and the boundaries of fault planes or unconformities) on the one hand and surface orientation (the gradient of the said isosurfaces) on the other hand. This last source of data allows the computation of a very smooth and continuous sedimentary structure, which is rarely the case in other freeware geostatistical tools (dell'Arciprete et al., 2012; Langousis et al., 2017). In
our case, the orientations (geological poles) are obtained by calculating the normal of the planes, defined by triangulation between the hydrofacies interfaces at the sampled data points.

Being specialized in geological modelling, GemPy handles the second-order (weak) stationarity of universal kriging by assuming a linear trend in the mean value of the scalar field. In addition, the random function defined for universal cokriging does not bear any physical meaning as it only aims at ensuring equality at every point of the isosurface (no matter the value).
As a result, the cross-variogram, inherent to cokriging, cannot be empirically determined. The shape of the surfaces mainly depends on the orientations provided and on an arbitrary spherical covariance function that only balances the relative weight of the surfaces and their orientation in the cokriging. Hence, the variogram parameters do not bear any physical meaning as well and are arbitrarily chosen to ensure stability to the computation according to the GemPy's developers' guidelines (De la Varga et al., 2019): the nugget effect should be small (set to 10 in our case) and the range equal to the domain's extension
(10,000 m in our case). As the variogram is not differentiated according to the search direction, the vertical component of the model must be exaggerated (x500 in our case) so that its dimension is compatible with the previously quoted values.

Finally, GemPy produces a 3D facies model made of 50 x 50 x 10 hexahedron elements. After rescaling on the z-direction, it has the same extension as the mesh used for the other models and a finer resolution. Henceforth, the facies in the flow/transport mesh for TRACES are determined according to the majority facies of the GemPy elements intersecting each 3D prismatic
element.

The calculation of both the scalar fields and their derivatives is handled by the Theano Python library, which also allows developments toward stochastic modelling. A more precise description of the software is available in De la Varga et al. (2019).

### 2.3.2 B-splines (deterministic interpolation)

Splines methods are also suitable for the construction of sedimentary models characterized by smooth surfaces. By definition,
their interpolation adjusts continuous polynomial equations to the data, ensuring no discontinuities and exact fitting (Prautzsch et al., 2002). Splines can be assimilated to flexible surfaces constrained to fit the observation values while minimizing their

bending energy. Contrary to simpler deterministic method (e.g. trend surface) that operate via a single polynomial equation, splines represent the surface in pieces and therefore require the computation of a large number of equations.

However, this method is chosen for its ability to reproduce smooth surfaces, compatible with a sedimentary morphology and can be easily carried through a GIS procedure (QGIS/SAGA multilevel b spline interpolation – Lee et al., 1997). To avoid anomalies in the stacking of the facies, the interpolation is carried on their thickness instead of their boundaries' z-coordinates. In addition, the first underlying facies is not interpolated but considered as the background (filling) lithology. The thicknesses of the four remaining facies are delivered in raster format, with integer values between 0 and 10 (i.e. the number of layers in the final 3D model), and a resolution of 200 m. Eventually, the facies stacking is transcribed for each column of prismatic elements in the 3D flow/transport mesh for TRACES according to the same majority analysis as for the GemPy procedure.

## 2.4 Hydrofacies parametrization and 3D simulations

The lithological models resulting from the interpolations do not have any assigned hydrodynamic parameters. Thus, an optimization procedure is implemented to find the hydraulic parameters of the five facies by minimizing the quadratic difference between 2D and 3D estimated transmissivities and porosities (Fig. 1, Eq. 7). Both previous steps of the methodology draw continuous data over the modelled domain (2D averaged parameters on the one hand and lithological structure on the other). Conceptually, the optimization could be performed with as many constraints as the number of elements of the mesh. However, the inversion and interpolation errors are expected to be minimal at the sampled data location. Therefore, the algorithm is carried out only with the parameter values and the lithological successions in these locations, minimizing uncertainties related to lack of sensitivity for transmissivity values or related to interpolation for lithological data.

The optimization is handled thanks to a Levenberg-Marquardt algorithm, whose unknowns are the hydraulic parameters (porosity and hydraulic conductivity) for each facies, i.e. 2×5 unknowns over the all domain. The constraints are the 2D mean values (transmissivity and porosity) at the sampled locations. In order to integrate the least amount of preconceptions in the method, the bounds of values within which the algorithm can pick during the optimization are not differentiated by facies (the bounds are $10^{-6}$ and $10^{-2}$ m.s$^{-1}$ for the hydraulic conductivity, 2 and 50 % for the porosity). The objective function of the optimization problem takes the form of Eq. 7.

$$O = \sum_i \left( \left( \sum_j l_{i,j} p_j - P_i \right)^{\mathrm{T}} \boldsymbol{\sigma}_i^{-1} \left( \sum_j l_{i,j} p_j - P_i \right) \right) \tag{7}$$

where $O$ is the objective function, $i$ is the index for the constraint (i.e. the sampled location retained for the optimization), $j$ is the index for each facies and $l_{i,j}$ is the thickness [m] of facies $j$ at location $i$. $p$ represents the parameters to be optimized (the hydraulic conductivity or the effective porosity of each facies) and $P$ the 2D mean values calibrated during the inversion stage, weighted by the matrix $\boldsymbol{\sigma}$ representing this calibration uncertainty. We consider only the diagonal of the matrix, containing the inverse of the variance given at location $i$ by the 2D calibration.

The final uncertainty of the optimized parameters is given by Eq. 8.

$$\epsilon_p = \varphi \left(\frac{\hat{O}}{m}\right)^{1/2} \left(\mathbf{C}_p\right)^{1/2} \tag{8}$$

where $\epsilon_p$ is the uncertainty of the parameter $p$, $\hat{O}$ is the objective function at end of the optimization and $m$ is the number of data. The coefficient $\varphi$ is determined through a Fisher's distribution, assuming a normal distribution of the uncertainty (for an estimation at 95 % of confidence, $\varphi = 1,96$). $\mathbf{C}_p$ is the variance of the parameter $p$, derived from the Jacobian (sensitivity matrix) of the model.

Once the optimization estimated each facies' hydraulic parameters, the 3D model is parametrized. Flow and contamination simulations are carried out with TRACES, as described previously, with the new facies distribution and the new parameter set. The boundary conditions and the recharge distribution are kept unchanged from the reference synthetic model. However, the initial state data are directly taken from the 2D simulations.

## 3 Results and discussion

### 3.1 Calibrated 2D models

The inversion algorithm is run 80 times for each sampling case to gather a set of possible solutions to the inverse problem, a model inversion lasting approximately 1 hr on average in these conditions. Each set of solutions is called a batch.

To avoid overparametrization, the adaptive refinement of the parameter grid, initially composed of 21 nodes (i.e., degrees of freedom for the minimization), is limited to three refinements. The number of new parameter located at the parameter grid vertices is also constrained by the number and location of the piezometric control points. Therefore, the final average number of parameter nodes is 497 for the dense sampling (400 control points) and 74 for the sparser sampling (40 control points).

Each solution batch produces very stable parameter fields (Tab. 3) and piezometer chronicles with a mean absolute discrepancy of less than 5 mm compared to the sampled reference data. For the sparse sampling, the mean absolute error increases to 37 cm when all control points from the dense sampling are included in the evaluation.

Table 3: Parameter variability between solutions in each batch

|  | Transmissivity relative standard deviation | | Porosity relative standard deviation | |
|---|---|---|---|---|
|  | Mean | Max | Mean | Max |
| Dense sampling | 0.5 % | 8.7 % | 1.0 % | 22.1 % |
| Sparse sampling | 0.2 % | 2.1 % | 1.5 % | 23.4 % |

The mean relative standard deviation of the parameter at the element mesh scale stays at a very low level in both cases. The variability of transmissivity is even lower for the sparse sampling (0.2 % vs. 0.5 %), as it has fewer degrees of freedom. The consistency between the estimated and the reference parameters is described in Sect. 3.3.

### 3.2 Three-dimensional interpolations

As the reference model is shaped according to a simple sedimentary pattern (absence of faults), both interpolation methods (geostatistical and deterministic) produce results of similar quality.

Differences in the facies composition of the models are marginal, even with a sparse distribution of the conditional data (Fig. 5).

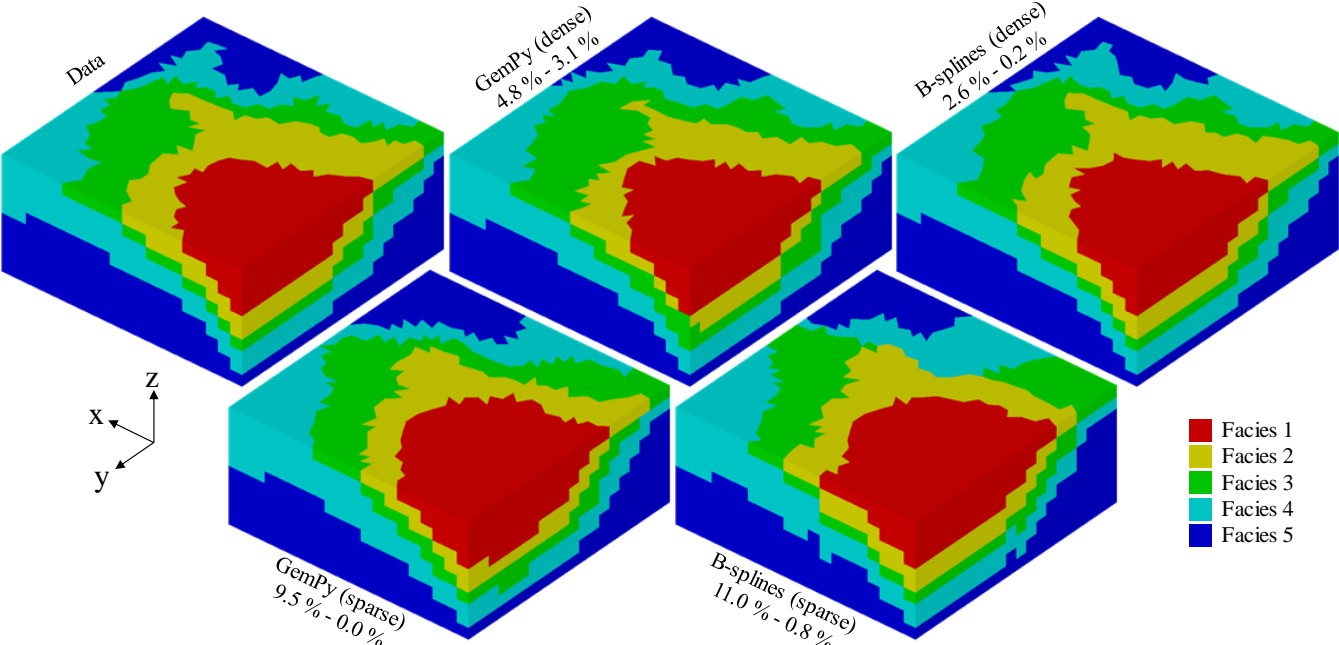

**Figure 5: Facies distribution of the models – the different facies are in greyscale; each model is shown with the ratio of misplaced**
**facies at the scale of the entire domain on one hand, and of the conditional data on the other hand.**

The two percentages accompanying each model in Fig. 5 represent the proportion of elements incorrectly parameterized for the whole data set and the conditional data, respectively. With a dense conditional data sampling, the deterministic approach yields slightly better results than the geostatistical one (2.6 % vs. 4.8 % of elements parametrized with the wrong facies). The GemPy algorithm handles sparser constraints slightly better (9.5 % vs 11.0 % of errors). However, the differences between the 350 two interpolation methods can be considered small and assumed to be dependent on the case study.

### 3.3 Parameter comparison

The results of the inversions (mean 2D values and their associated variance) and the known facies distributions at the sampling location are used to optimize 3D hydrodynamic parameters as explained in section 2.4. These optimized values and their uncertainties are shown in Fig. 6.

In all cases, the hydraulic conductivity of each facies stays in the same order of magnitude as the reference data (the mean errors account for 26 % of the known hydraulic conductivity values). The largest discrepancy (0.26 log unit, 45.5 % of the reference value in a linear scale) affects the facies 4 in the sparse sampling-based estimation. The gaps between the data and the estimated porosity are also very low, with mean and maximal absolute errors of 1.3 % and 2.9 % (i.e., 2.9 % and 6.4 % of the data values). These consistent results are due to the fact that the parametrization optimization is carried out with input at

the sampled data location, where the 3D facies vertical succession is known and where the errors on the 2D mean parameters are low.

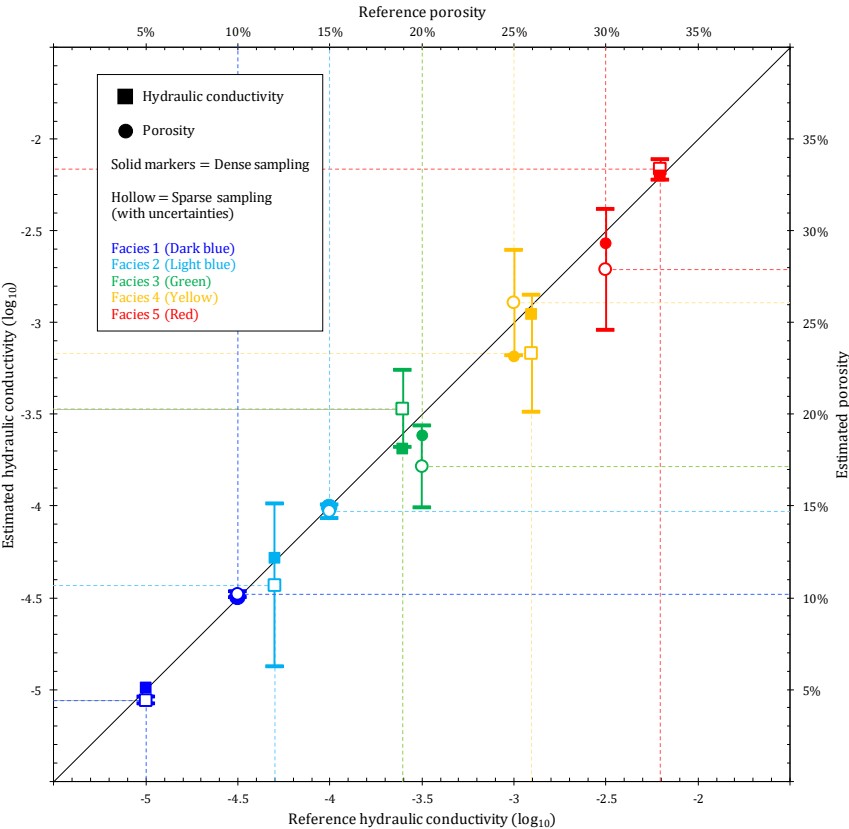

**Figure 6: Calibrated facies parameter values and uncertainties – on the x-axes, the reference parameter values, on the y-axes, the estimated values; the facies are differentiated by colours, the types of parameter are differentiated by markers (hollow markers**
**are for sparse sampling); 95 % confidence intervals are only reported for the sparse models.**

The uncertainty attached to the optimization varies with the density of the data. The values related to the dense sampling are negligible (the highest uncertainties are at 0.02 log unit for the hydraulic conductivity and 0.5 % for the porosity), therefore only the ones of the sparse sampling are shown. At the most, the uncertainty extends over 0.44 log units for the hydraulic conductivity (facies 2) and 3.3 % for the porosity (facies 5). Consequently, the confidence intervals almost always include the corresponding reference value.

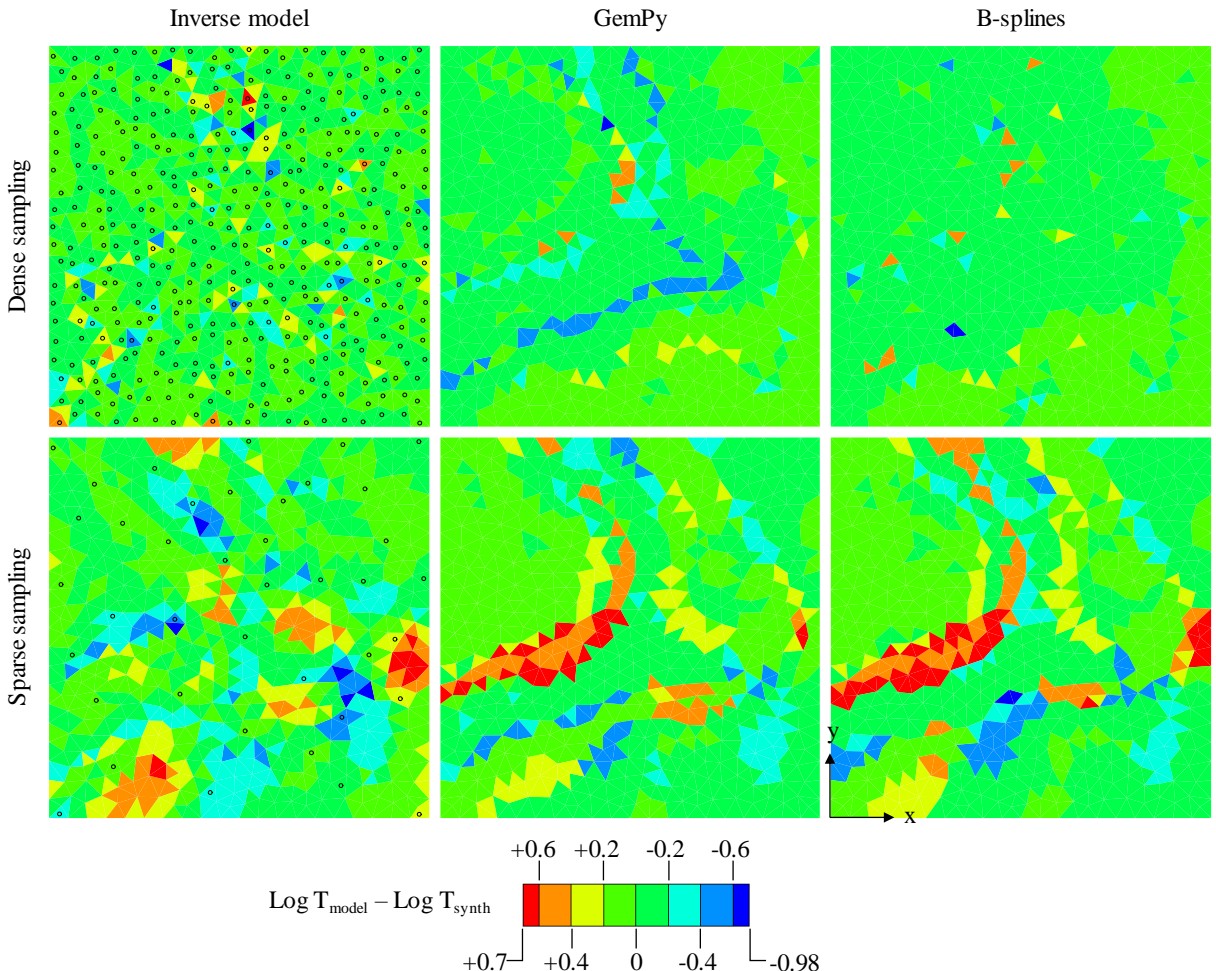

Figure 7: Map of transmissivity discrepancies – the values represent the difference between the models output and the initial synthetic data; the samplings (dots) are underlined on the 2D calibrated (inverse) models.

The conjunction of the parameter discrepancies and facies misplacements results in transmissivity errors, as shown in Fig. 7. The transmissivity discrepancies are always below one order of magnitude and mostly below 0.2 log units. The large-scale heterogeneities are very well reproduced in every case, notably in the models based on sparse data.

Comparisons between the 2D parameters and the 3D parameters show that the estimation errors in the former do not propagate entirely in the latter. Indeed, the errors of the inverse models are not only located at the interfaces between the large-scale

heterogeneities but also at smaller scales, within the heterogeneities, when the density of the constraint data is reduced (in the absence of local piezometric data, the hydrodynamic parameters are less constrained; see bottom left of Fig. 7). The 3D interpolation techniques are not subject to these small-scale errors as they generate very smooth and continuous facies distributions (Fig. 5). Therefore, the final discrepancies are mainly at the transitions between the large-scale horizontal heterogeneities where the facies interpolation generates errors (in particular, the overestimation of transmissivity is visible

where the interpolations have extended facies 1 in excess).

### 3.4 Piezometric heads comparison

In view of the few parameterization errors produced with the dense sampling, the results with respect to piezometry are shown only for the sparse sampling. In addition to the simulations incorporating the optimized hydrodynamic parameters per facies, 4 additional runs are conducted for each interpolation method in order to study the propagation of parameter uncertainties. For

each additional runs, the parameter values are set: (i) at the upper bound of the confidence interval, (ii) at the lower bound, (iii) alternately at the lower (for the facies 1,3,5) and upper (facies 2,4) bounds, (iv) alternately at the lower (for the facies 2,4) and upper (facies 1,3,5) bounds of the parameter confidence interval. These simulations generate chronicles and maps whose extreme values are retained to construct "envelope curves", showing the final uncertainty of the piezometry (Fig. 8).

The final piezometric state for each model (water head averaged on the vertical) is shown in the central map of Fig. 8. The

piezometric contours are well reproduced, especially for the 40 m and 30 m isolines. Differences between the interpolation strategies are marginal. The main discrepancy between the models and the reference data is visible in the middle of the right border of the domain, where the interpolations underestimated the presence of facies 1, in favour of facies 2. However, the narrow confidence interval on the parameters of facies 1 is reflected by a low uncertainty on the piezometry in the lower right corner of the domain, where this facies predominates. In contrast, the uncertainty in piezometry is significant where facies 2,

3 and 4 predominate.

The water head fluctuations are also consistent with the reference chronicles over the entire period of simulation and for every recharge zone. Amongst the chosen piezometers, those identified with an asterisk were not used as constraint data in the sparse sampling. The mean absolute errors of each model are 44.6 cm for the GemPy model and 47.6 cm for the B-splines. The deviations mainly take the form of a shift (by excess or by default) in the base level when the fluctuations are well reproduced.

Overall, the parametrization discrepancies (Fig. 6) are too small to significantly modify the flow dynamics. Combined with the small differences in the model's composition (Fig. 5), the water head equilibrium is slightly shifted as shown in the charts. Therefore, a more significant deviation would occur in a permanent flow simulation (with averaged recharge). As with the contour map, the highest levels of uncertainty are for piezometers intercepting significant thicknesses of facies 2 (P2, P3, P6 and P7), this one having the widest confidence intervals.

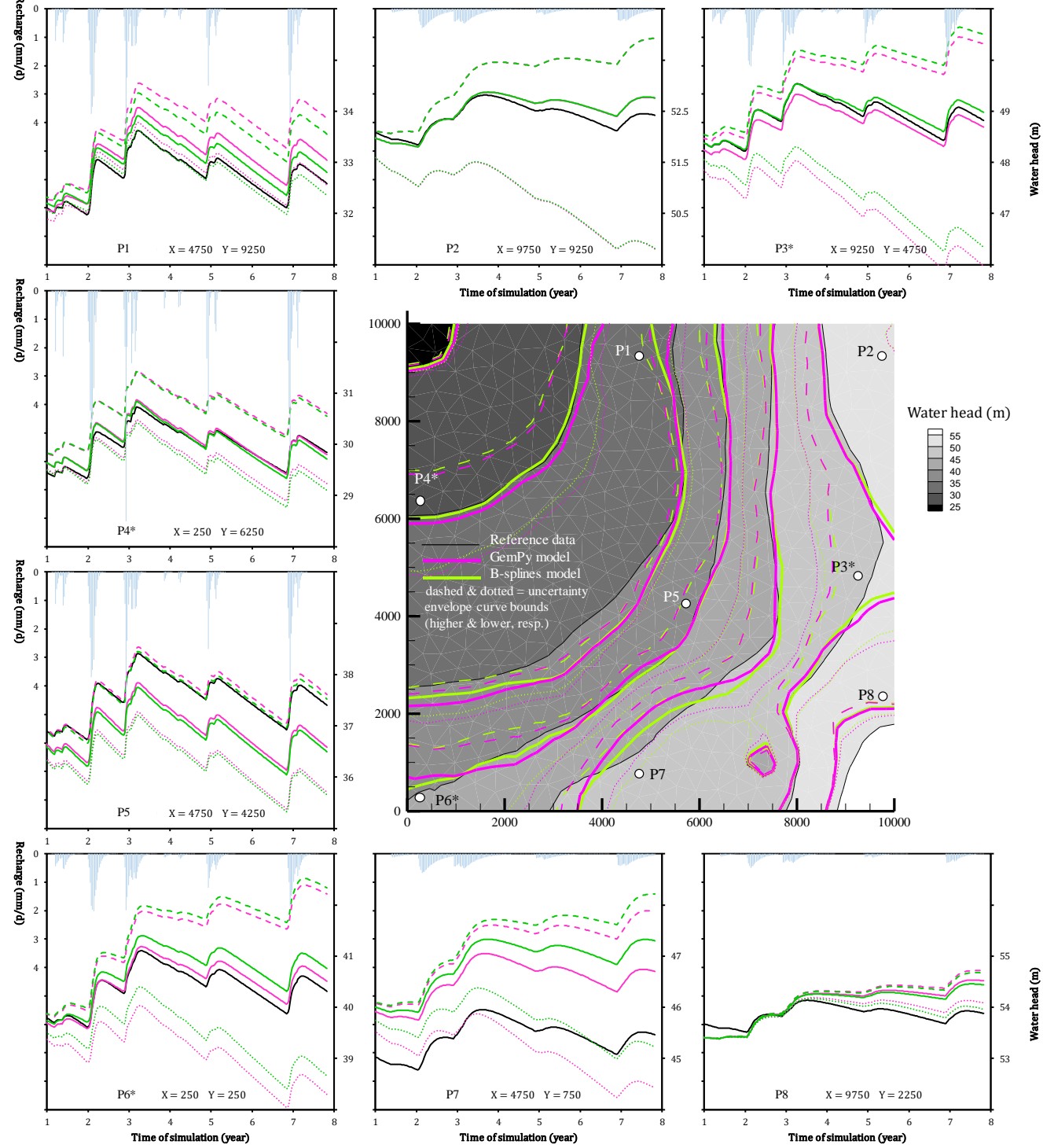


**Figure 8: Comparisons of the 3D piezometric heads variations – piezometric heads values (chronicles and map) are differentiated by colours (black: reference model, purple: GemPy, green: B-splines); the dashed and dotted lines give the uncertainty intervals.**

**3.5 Contamination comparison**

The study of contaminant transport is another prime use of hydrogeological models. The outputs of the transport simulations are shown in Fig. 9, in the form of breakthrough curves and iso-concentration maps (10 mg.l$^{-1}$) at different times of the simulation. As for the piezometric data, envelope curves of uncertainty are deduced from the 4 simulations involving the parameters at the bounds of the confidence intervals.

The models' output errors are attributable to parameterization discrepancy on individual facies, interpolation misplacements and the accumulation of these same errors upstream of each surveyed location. The results confirm that contaminant transport is much more sensitive to parameterization errors than piezometry. Indeed, the latter is governed by the transmissivity, where individual facies misparametrization can be buffered by the vertical integration. For contaminant transport, each voxel parametrization may influence the outcome.

For instance, source 1 is located on facies 1 in the reference model and on facies 2 in the interpolated models. Therefore, the dynamics of the breakthrough curve and the pollutant plume are significantly different (i.e. a lower spike due to a higher porosity, and a faster depletion due to a higher conductivity). On the contrary, the facies distribution is preserved in the location of source S2 (in facies 4), where the discrepancies are mainly due to hydrodynamic parametrization errors on this very same facies.

Overall, the results at the outlet E and the plumes maps show that, in our case, the parametrization errors have a more significant impact on the fate of sources S2 and S3 than for source 1.

The confidence intervals on the facies parameters (especially on facies 2, 3 and 4) generate scenarios where the plumes disappear early on one hand or extend in excess on the other hand.

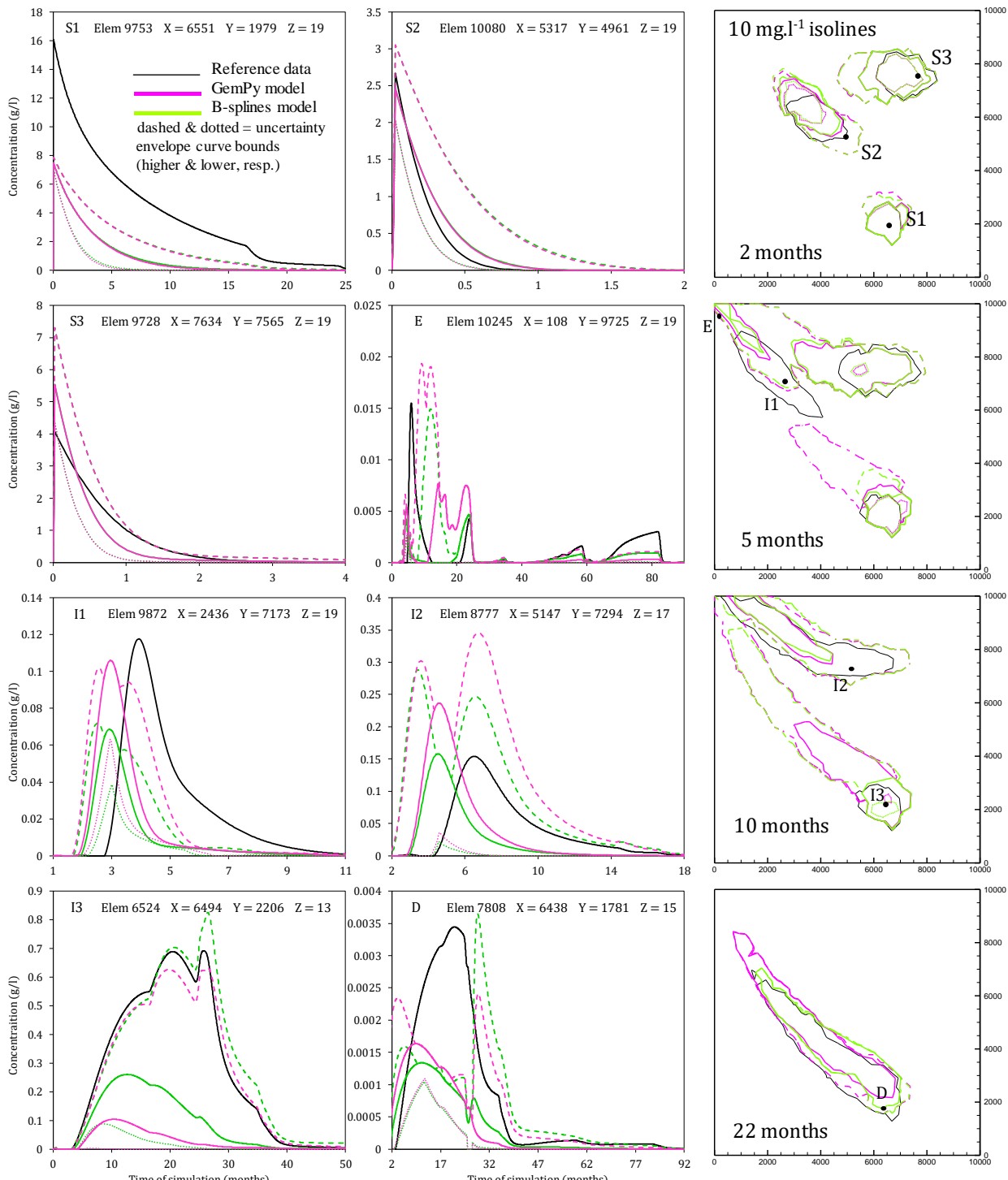

**Figure 9: Contamination breakthrough curves – GemPy (green) and B-splines (yellow) parametrized models (dashed for sparse data) are compared to the initial synthetic dataset (black); the two rows on the left show depletion or breakthrough curves, the last row on the right depicts contamination plumes at four different times.**


### 3.6 Perspectives regarding uncertainties

In this study, only the uncertainty related to the calibration of the 2D parameters is calculated and propagated to the 3D models. However, it must be emphasized that in the context of distributed hydrogeological models, many other sources of uncertainty occur (Pechlivanidis et al, 2011).

First, there are data uncertainties: piezometric measurements, rainfall and radiative data (leading to recharge estimates), lithological descriptions (both in terms of their categorization and altimetry) are all subject to error. Our approach via a synthetic case encouraged us to postpone this aspect and to concentrate on the analysis of the methodology. These uncertainties will be taken into account in future work dealing with real cases.

Second, all uncertainties related to the parameters have not been addressed in this study. Focused on the determination of

hydrodynamic parameters, it did not integrate the uncertainty related to the dispersivity and molecular diffusion parameters inherent to the contaminant transport phenomena.

Third, the structure of the models and the way they are defined also involve uncertainties. In particular, interpolation methods produce uncertainty and propagate those inherent in the lithological data (Phillips & Mark, 1996; Lloyd & Atkison, 2001; White, 2017). Our data sampling strategy allowed to circumvent this pitfall, but it becomes unavoidable when there are

significant gaps in the data.

Monte Carlo simulations (Wagener & Collat, 2007; Beven, 2009) allow the joint estimation of these uncertainties, at great computational cost (the simulations must cover a range of value for each input or parameter at stake). Several random (Olsson & Sandberg, 2002) or Bayesian (Vrugt et al., 2009) sampling stategies are at hand in order to reduce the number of iteration needed to obtain a representative view of the uncertainties.

Integrating this type of analysis in the framework of our method is one the important improvements planned.

## 4 Conclusions

A method has been developed to assess 3D aquifer parameters by combining t hydrodynamic parameters estimated by a 2D model calibration and 3D facies interpolation. While direct 3D parameter estimation is generally based on a heavy geophysical survey, the proposed methodology is based solely on piezometric series and geological logs (commonly available at the same

locations). Both 2D flow model calibration and 3D interpolation parts of the algorithm are independent. Therefore, the approach is not restricted to the tools described in the article (i.e., other interpolation methods than GemPy and B-splines can be used) and can potentially incorporate a pre-existing 2D model.

The synthetic test carried with a relatively sparse dataset yields a consistent hydrodynamic parametrization (highest discrepancy: 45.5 % of the initial value for hydraulic conductivity, 6.4 % for effective porosity) and quite low errors in facies

distribution (11 % of misplaced facies at the most). Subsequently, the reconstructed piezometric series show very consistent dynamics with a maximal mean difference of 47.6 cm, mainly due to shifting the base level, while the fluctuations and the

hydraulic gradients are generally unaltered. The discrepancies concerning the transport simulations are more significant, the phenomenon being more sensitive to parameterization errors at the individual voxel scale.

Comparatively to joint inversion methods, the need of data acquisition and the computation efforts are lower. However, in a
field context, the method is very dependent on the characterization of the hydrofacies and the quality of the piezometric survey. Indeed, if uncertainties related to the 2D flow calibration were propagated to the 3D parameter optimization and, thereafter, into the piezometry and contaminant transport simulation, other sources of uncertainty hindering the hydrogeological modelling process were not accounted for at the time of publication. Among them, we can cite the uncertainties related to input data, transport parameters or the structure of the model itself (especially the lithology interpolation step).

Another pitfall of the method, inherent to the 2D step, lies in the fact that low-hydraulic conductivity facies may be masked by more permeable facies in the transmissivity term, making their parameterization somehow difficult. Also, some other important modelling points are not addressed in the study, e.g. the vadose zone dynamics and the transport parameters, that require separate estimates. Following this synthetic case, we plan to test the method on a real case, in order to confirm its operational potential, completed by comprehensive sensitivity and uncertainty analyses.

**Code and data availability**

TRACES, PINOGRI and the synthetic case data set can be provided on request to the reference author.

GemPy is an open-source 3D geomodelling package (www.gempy.org).

The B-splines interpolation method used is an add-in tool of the free and open source geographic information system QGIS/SAGA.

**Author contribution**

D. Rambourg and P. Ackerer designed the method workflow, D. Rambourg designed the synthetic case and carried out the simulations. R. Di Chiara adapted the GemPy Python package, P. Ackerer developed TRACES (with co-authors cited in References) and co-developed PINOGRI with D. Rambourg (and previous contributors to the code development, as cited in References). D. Rambourg prepared the manuscript with contributions from all the co-authors.

**Competing interest**

The authors declare that they have no conflict of interest.

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
