# Peer review of "3D hydrogeological parametrization using sparse piezometric data"

_EGUsphere, 2022_

## Referee Comment (RC2)

**3D hydrogeological parametrization using sparse piezometric data**

D. Rambourg, R. Di Chiara and P. Ackerer

The manuscript is well structured and well written.

In my opinion, the main novelty of the manuscript is the following: estimation of hydrodynamic characteristics ($k$ and $n$) of a 3D flow model comparing the calibrated 2D transmissivities rather than the hydraulic head measurements.

At the beginning, the transmissivities of a 2D flow model are estimated comparing sparse measurements of hydraulic heads obtained by means of piezometers (actually the aquifer in this study is synthetic) with the heads calculated by the 2D flow model. According to vertical logs data collected in the piezometers a 3D reconstruction of litho-facies is obtained and e 3D flow model realized. In order to estimate the hydraulic conductivity values for each element of the 3D model an inverse procedure involving the transmissivities is implemented: the hydraulic conductivity in each facies is calculated optimizing the distance between the 2D inversion transmissivities and the 3D transmissivities.

In literature already exist studies in which the aquifer is conceptualized as a multiple-continuum, where the volumetric fraction of a geo-material within a cell of the numerical flow model is calculated by Multiple Indicator Kriging and the hydraulic head data are embedded jointly within a three-dimensional inverse model of groundwater flow: model parameters ($k$ and $n$) are estimate by a Maximum Likelihood fit between measured and modeled - vertically average - hydraulic heads, resulting in a spatially heterogeneous distribution of hydraulic conductivity (Guadagnini et al., 2004; Straface et al., 2011).

The authors should support their approach, i.e., the transmissivities versus the hydraulic heads conditioning, comparing the two inversion strategies and showing the advantage to compare the 2D transmissivities rather than the vertically averaged hydraulic heads.

I haven't any concerns about the mathematical description of the equations presented herein and the inversion procedures, nevertheless I recommend the above mentioned comparison before its publication.

**References**

Guadagnini L., Guadagnini A, Tartakovsky DM (2004). *Probabilistic Reconstruction of geologic facies.* J. of Hydrol., 294, 57-67.

Straface S., Chidichimo F., Rizzo E., Riva M., Barrash W., Revil A.,Cardiff M., Guadagnini A. *Joint inversion of steady-state hydrologic and self-potential data for 3D hydraulic conductivity distribution at the Boise Hydrogeophysical Research Site.* Journal of Hydrology, 2011, Vol. 407, pp. 115–128.

---

## Author Comment (AC1)

**3D hydrogeological parametrization using sparse piezometric data**

*D. Rambourg, R. Di Chiara and P. Ackerer*

Dear Referee, please find below the point by point answers (in blue) to your comments (in black).
* * *
Dear authors,

I read carefully your work and I found it a very important contribution in groundwater hydrology.

The methodology is clearly presented, the mathematical background as well. The interpolation part may require some more details but on the other hand it mainly supports the concept

> We thank the reviewer for his very positive appreciation of our work.
>
> We agree that the interpolation details are not crucial for the understanding of the methodology. However, we will provide more information about the main ideas embedded in GemPy and the B-splines method in the revised version.

1) The proposed method is only digestive for those who have specialized knowledge of the entire tools presented.

> We regret that the description is not as clear as we would like. In the end, the process is relatively straightforward:
>
> 1a) Piezometric series are used to invert 2D groundwater flow parameters (i.e. conductivity and effective porosity)
>
> 1b) Drilling logs are used to interpolate 3D lithological distribution
>
> 2) Flow parameters of each classified lithology is optimized in order to make the final 3D model and the 2D inversion fit locally in terms of transmissivity and porosity.
>
> We'll add further description about each of these steps.

2) Please mention the innovation compared to similar works.

> Similar approaches are very few in the literature.
>
> Harp et al. (2008) also used a combination of methods in order to identify aquifer structure, namely a 2D inversion procedure based on minimization of the residuals of hydraulic heads for the flow parameters and a transition probability model for the aquifer structure. Two main points make our study innovative and more advanced compared to it. We tested our method on 5 facies while Harp et al. only designed their aquifer with 2 facies. And speaking of design, their synthetic case is produced with the same tool (T-PROGS) that is then used for

reconstructing the aquifer structure. In our case, the initial structure was designed by hand, independently of the interpolation tools used afterwards.

The approach of Viaroli et al. (2019) is less similar to ours. They used a 2D simplification to assess boundary conditions and recharge of an already developed 3D model for a real case application.

We'll integrate these comparisons with the cited literature in the introduction (lines 33-34).

3) Most important the presented methodology is very complex to be reproduced. I am not saying that is bad! but there also similar works in the literature that do the same work with a simpler manner. Maybe it would be good, if possible, to have a comparison with one of them. Your method is more detailed but compared to simpler approach the performance is far more efficient? Otherwise, what is the reason to have such many methodological and sometime complex steps.

First of all, our idea is motivated by the usual way of groundwater calibration which is mostly performed in 2 dimensions. We question the use of the transmissivity field obtained through 2D calibration to build a 3D representation of the aquifer i.e. to estimate the structure of the heterogeneities and their related hydraulic conductivity. Our approach consists in estimating the transmissivity first with a 2D model calibration and then to build the 3D representation of the aquifer using the calibrated transmissivities only. It is quite different from the approaches dealing with 3D representation of an aquifer by joint inversion. We do believe that this approach is simpler than the other ones. It can also take benefit of an existing already calibrated 2D model to build a 3D transport model.

Moreover, the method is not so complex considering what it offers: estimating both 3D flow parameters and aquifer structures, based solely on piezometric series and vertical logs descriptions. Most studies on 3D aquifer modeling use multi-methods/joint inversions based on costly geophysical field data. We do not have experience of those studies that propose 3D modeling in an even simpler way.

4) paragraph 2.4 The optimization part needs more details. It is not clear how optimization works here.

At a given location of the domain, the optimization is performed using:

- A transmissivity value estimated by the calibration of the 2D model;
- A vertical distribution of a given number (n) of different porous materials (facies) with their related thickness obtained by facies interpolation.

The unknowns of the optimization process are the hydraulic parameters (porosity, hydraulic conductivity) for each facies, i.e., 2n unknowns over the all domain. The constraints are the transmissivity values estimated at each selected cell of the 2D groundwater flow model. Estimating the hydraulic properties face an overdetermined problem, the number of unknowns is much less than the number of constraints. We solve this problem through optimization and use only the most relevant values of transmissivity, i.e., the less uncertain. These values of transmissivities are located at wells where measurement exists, either piezometric heads or lithological data. Therefore, uncertainties related to lack of sensitivity for transmissivity values or related to interpolation for lithological data are minimized.

More details following the previous paragraph will be provided in the revised version to make it more understandable.

5)The proposed 3d methodology consist of inversion, interpolation, optimization. All these steps consider parameters. Therefore, an uncertainty analysis is required to study the uncertainty propagation.

We agree that an uncertainty analysis is a critical step in a modelling process. However, in the article, the choice is made to describe a new method through a synthetic case, independently of any data uncertainty. We focused on the effect of data sparsity to assess the method applicability. As uncertainty analysis does not only depend on the methods implemented, but also on the structure of the model, it appears to be more relevant in real case application, where the data is indeed subject to uncertainty and where the resulting level of uncertainty in the model outputs has an operational impact. We plan to implement this kind of analysis in the next development of the tool.

6) How realistic is the upscale of such model to a real case study. I understand the research orientation which is very strong but, this is also a matter of discussion.

A first test in real case was initiated in parallel with the development of the method, showing its applicative potential but also highlighting its dependence to a strong lithological classification (matter already discussed in the conclusion). In addition, the 2D step means that low-permeability facies may be masked by more permeable facies in the transmissivity term, making their parameterization sometimes difficult. And lastly, some other points are not addressed in the study, e.g. the vadose zone and the transport parameters, which require separate estimates.

These points will be added in the conclusion.

7) The fixed parameters of the aquifer model regarding transport it would be good to be accompanied by a sensitivity analysis.

Our choice not to perform a sensitivity analysis follows the same logic as for the uncertainty analysis. In this manuscript, we test the feasibility of this new methodology. We show that it is promising and uncertainties will be addressed in a further paper dedicated to the reliability of 3D transport modelling using this approach to build the 3D model.

---

## Author Comment (AC2)

**3D hydrogeological parametrization using sparse piezometric data**

*D. Rambourg, R. Di Chiara and P. Ackerer*

Dear Referee, please find below the point by point answers (in blue) to your comments (in black).
* * *
The manuscript is well structured and well written.

In my opinion, the main novelty of the manuscript is the following: estimation of hydrodynamic characteristics (k and n) of a 3D flow model comparing the calibrated 2D transmissivities rather than the hydraulic head measurements. At the beginning, the transmissivities of a 2D flow model are estimated comparing sparse measurements of hydraulic heads obtained by means of piezometers (actually the aquifer in this study is synthetic) with the heads calculated by the 2D flow model. According to vertical logs data collected in the piezometers a 3D reconstruction of litho-facies is obtained and e 3D flow model realized. In order to estimate the hydraulic conductivity values for each element of the 3D model an inverse procedure involving the transmissivities is implemented: the hydraulic conductivity in each facies is calculated optimizing the distance between the 2D inversion transmissivities and the 3D transmissivities.

> We thank the reviewer for his appreciation of the work, and we are glad that it can be understood in the way that this summary demonstrates.

In literature already exist studies in which the aquifer is conceptualized as a multiple-continuum, where the volumetric fraction of a geo-material within a cell of the numerical flow model is calculated by Multiple Indicator Kriging and the hydraulic head data are embedded jointly within a three-dimensional inverse model of groundwater flow: model parameters (k and n) are estimate by a Maximum Likelihood fit between measured and modeled - vertically average - hydraulic heads, resulting in a spatially heterogeneous distribution of hydraulic conductivity (Guadagnini et al., 2004; Straface et al., 2011).

The authors should support their approach, i.e., the transmissivities versus the hydraulic heads conditioning, comparing the two inversion strategies and showing the advantage to compare the 2D transmissivities rather than the vertically averaged hydraulic heads.

> We also thank the reviewer for his comments that give us the opportunity to better highlight the innovative aspects or our work.
>
> First of all, our idea is motivated by the usual way of groundwater calibration which is mostly performed in 2 dimensions. We question the use of the transmissivity field obtained through 2D calibration to build a 3D representation of the aquifer i.e. to estimate the structure of the heterogeneities and their related hydraulic conductivity. Our approach consists in estimating the transmissivity first with a 2D model calibration and then to build the 3D representation of the aquifer using the calibrated transmissivities only. It is quite different from the approaches dealing with 3D representation of an aquifer by joint inversion of 3D hydraulic data conditioned by the piezometric heads or conditioned by piezometric heads and geophysical

data (Straface et al., 2011) or measured parameter values through direct measurements or indicators (Guadagnini et al., 2004.).

The choice of using a 2D model inversion, leading to transmissivity conditioning, instead of a 3D inversion, is also guided by 2 criteria.

First, the hydraulic heads are little sensitive to the vertical heterogeneity. Therefore, their use to calibrate hydraulic conductivities without additional constraints is more pertinent in 2D.

Second, 2D calibrations are way more parsimonious than the 3D ones in terms of data management and computational effort.

We will add some sentences in the introduction to better highlight these innovative aspects compared to other ones.

---

## Author Response (AR1)

**3D hydrogeological parametrization using sparse piezometric data**

*D. Rambourg, R. Di Chiara and P. Ackerer*

Dear Referee, the additions are highlight in yellow in the track-changes version of the manuscript uploaded in Egusphere. We list below the changes corresponding to each of yours remarks.
* * *
RC#1

Dear authors,

1) The proposed method is only digestive for those who have specialized knowledge of the entire tools presented.

> Both more general introduction and more precise elements on the tools were added.
>
> For TRACES and direct calculations: lines 106-107; 140-141; 144-146
>
> For PINOGRI and inversion: lines 170-173; 197-201; 215-216
>
> For facies interpolation (GemPy and B-splines): lines 228-230; 236-240; 259-262
>
> For the optimization of the hydrofacies parameters: lines 274-286

2) Please mention the innovation compared to similar works.

> As stated in our previous response, we integrated comparison with Viaroli et al. (2019) (lines 36-38) and Harp et al. (2008) (lines 45-47).

3) Most important the presented methodology is very complex to be reproduced. I am not saying that is bad! but there also similar works in the literature that do the same work with a simpler manner. Maybe it would be good, if possible, to have a comparison with one of them. Your method is more detailed but compared to simpler approach the performance is far more efficient? Otherwise, what is the reason to have such many methodological and sometime complex steps.

> Along with the response given in a previous round of reviewing, we didn't add elements regarding this point. Indeed, our method is not so complex considering what it offers: estimating both 3D flow parameters and aquifer structures, based solely on piezometric series and vertical logs descriptions. Most studies on 3D aquifer modeling use multi-methods/joint inversions based on costly geophysical field data. We do not have experience of those studies that propose 3D modeling in an even simpler way.

4) paragraph 2.4 The optimization part needs more details. It is not clear how optimization works here.

*More details, including an equation formulation of the optimization problem, have been added on lines 274-286.*

5)The proposed 3d methodology consist of inversion, interpolation, optimization. All these steps consider parameters. Therefore, an uncertainty analysis is required to study the uncertainty propagation.

6) How realistic is the upscale of such model to a real case study. I understand the research orientation which is very strong but, this is also a matter of discussion.

7) The fixed parameters of the aquifer model regarding transport it would be good to be accompanied by a sensitivity analysis.

*These last 3 points were addressed in the conclusion (lines 403-407).*
* * *
RC#2

The authors should support their approach, i.e., the transmissivities versus the hydraulic heads conditioning, comparing the two inversion strategies and showing the advantage to compare the 2D transmissivities rather than the vertically averaged hydraulic heads.

*The advantage of our method relying on a 2D transmissivity constraint, both in terms of data acquisition and computational efforts, compared to 3D joint inversions, is highlighted in the introduction (lines 37-43). The references provided by the referee (Straface et al. 2011; Guadagnini et al. 2004) have been added (lines 33-35) to complete the state-of-the art and to introduce the novelty of our own approach.*

---

## Author Response (AR2)

**3D hydrogeological parametrization using sparse piezometric data**

*D. Rambourg, R. Di Chiara and P. Ackerer*

Dear Referee, the additions/modifications are highlighted in yellow in the track-changes version of the manuscript uploaded in Egusphere. We list below the changes corresponding to each of your remarks.
* * *
Dear authors I believe your manuscripts is a useful input but reading it I have the following concerns for clarifications and potential improvements.

We thank you greatly for your remarks and your input that really benefited our work.

The interpolation and geostatistical part needs improvement in terms of variogram calculation, stationary analysis, parameters calculation, validation and variance-uncertainty. It is a major part of your methodology and needs to be clear. You mention application of universal cokriging, of which variables, using which variogram model e.t.c

GemPy performs a cokriging of two types of data: isosurface (i.e. the interface between stacked lithologies) and the interface orientation. It uses an arbitrary spherical covariance function that only balances the relative weight of the surfaces and their orientation in the cokriging, as the random function defined in GemPy's method does not bear any physical meaning and is dimensionless. It only aims at ensuring equality at every point of the isosurface.

We propose to add the following information to this section [in brackets, the text was already in the previous version of the manuscript]:

*[As a result, the cross-variogram, inherent to cokriging, cannot be empirically determined. The shape of the surfaces mainly depends on the orientations provided and on an arbitrary spherical covariance function that only balances the relative weight of the surfaces and their orientation in the cokriging.] Hence, the variogram parameters do not bear any physical meaning as well and are arbitrarily chosen to ensure stability to the computation according to the GemPy's developers' guidelines (De la Varga et al., 2019): the nugget effect should be small (set to 10 in our case) and the range equal to the domain's extension (10,000 m in our case). As the variogram is not differentiated according to the search direction, the vertical component of the model must be exaggerated (x500 in our case) so that its dimension is compatible with the previously quoted values.*

*Finally, GemPy produces a 3D facies model made of 50 x 50 x 10 hexahedron elements. After rescaling on the z-direction, it has the same extension as the mesh used for the other models and a finer resolution. Henceforth, the facies in the flow/transport mesh for TRACES are determined according to the majority facies of the GemPy elements intersecting each 3D prismatic element.*

it seems that an uncertainty analysis has been carried out in line 280 but it is not clearly presented and discussed. As long as simulations have been carried out then the overall uncertainty can be presented in the facies of the models as support figures.

The uncertainty attached to the optimization step is now introduced in the methodology:

*The objective function of the optimization problem takes the form of Eq. 7.*

$$O = \sum_i \left( \left( \sum_j l_{i,j} p_j - P_i \right)^T \boldsymbol{\sigma}_i^{-1} \left( \sum_j l_{i,j} p_j - P_i \right) \right) \qquad (7)$$

*where $O$ is the objective function, $i$ is the index for the constraint (i.e. the sampled location retained for the optimization), $j$ is the index for each facies and $l_{i,j}$ is the thickness [m] of facies $j$ at location $i$. $p$ represents the parameters to be optimized (the hydraulic conductivity or the effective porosity of each facies) and $P$ the 2D mean values calibrated during the inversion stage, weighted by the matrix $\boldsymbol{\sigma}$ representing this calibration uncertainty. We consider only the diagonal of the matrix, containing the inverse of the variance given at location $i$ by the 2D calibration.*

*The final uncertainty of the optimized parameters is given by Eq. 8.*

$$\epsilon_p = \varphi \left( \frac{\hat{O}}{m} \right)^{1/2} \left( \boldsymbol{C}_p \right)^{1/2} \qquad (8)$$

*where $\epsilon_p$ is the uncertainty of the parameter $p$, $\hat{O}$ is the objective function at end of the optimization and $m$ is the number of data. The coefficient $\varphi$ is determined through a Fisher's distribution, assuming a normal distribution of the uncertainty (for an estimation at 95 % of confidence, $\varphi = 1,96$). $\boldsymbol{C}_p$ is the variance of the parameter $p$, derived from the Jacobian (sensitivity matrix) of the model.*

The integration of the variance of the 2D parameters slightly modifies the 3D optimized values and allows to give confidence intervals on the final facies parameters (see Sections 3.3 and following).

Your method needs to be compared with a classical 3d inverse modelling approach (3D inverse modelling of groundwater) such the ones that have been published by J. J. Gómez-Hernández and Kitanidis.

The work of J. J. Gómez-Hernández and Kitanidis is added in the introduction, along with the work of Straface and Guadagnini, to better stress the difference with classical 3D inverse modelling approach, that still rely on more heavy and specific data collection:

*[But the latter is less sensitive to the vertical structure of the aquifer, leaving its estimation dependent on complex and expensive field methods – e.g. pumping tests (De Caro et al., 2020), tracer tests (Linde et al., 2006), electrical resistivity (Coscia et al., 2011; Priyanka & Mohan Kumar, 2019), radar tomography (Boni et al., 2020), self-potential methods (Eppelbaum, 2021), crosshole testing (Klotzsche et al., 2013; Doetsch et al., 2010), hydraulic tomography (Sanchez-León et al., 2015; Luo et al., 2020; Fischer et al., 2020) – and/or laboratory analysis – e.g. grain-size analysis from core samples (Marini et al., 2018) and ex-situ permeability tests (Zhang & Brusseau, 2005).] The collection of these information, describing the vertical heterogeneity of the aquifer, allows the development of 3D inversion techniques. For example, some successful methods combine direct parameter quantification and stochastic geological modelling (Guadagnini et al., 2004; Fu & Gómez-Hernández, 2008; Cardiff & Kitanidis, 2009), others incorporate water head data and more advanced geophysical measurements to the (joint) inversion procedure (Straface et al., 2011; Lee & Kitanidis, 2014).*

In the abstract it is mentioned: Finally, the parameters of each facies (hydraulic conductivity and porosity) are obtained through an optimization loop, that minimizes the difference between the 2D calibrated transmissivity and the transmissivity computed with the estimated 3D facies parameters.

According to presented methodology to achieve the target inversion, interpolation, optimization is applied. All these steps consider parameters. Therefore, an uncertainty analysis is required to study the uncertainty propagation even for synthetic data or at least show the overall uncertainty thorough the simulations, or even identify the sources of uncertainty for the readers to consider them when up-scaling.

Uncertainties concerning the 2D inversion (i.e. the dispersion of transmissivities amongst the different solutions) is now propagated to the optimization (i.e. uncertainty of the facies parameters).

Uncertainties concerning the interpolation has been circumvented by integrating in the parameter optimization only location where the lithology is known.

In addition, the uncertainty occurring when classifying the facies is mentioned but not addressed in this paper. As well, transport parameters are not estimated, as stated in the conclusion, it should be the subject of further research.

Also, a note has been added to make the reader aware of the existence of uncertainty in the water head data (in Section 2.2.):

*Although piezometric data is subject to uncertainty in a field context, we do not address this aspect in the present study and the water heads measurements errors are considered as negligible.*

The need of comprehensive uncertainty and sensitivity analyses is outlined in the conclusion.

for the B-splines method you mention that one can use GIS, for interpolation python and for the model fortan!
How all these software results are connected for your complete model? It seems that many assumptions should be made and I am still worried about uncertainty.

More information about the bridges between the different tools is added. Concerning GemPy, see above (response to the first comment).

Concerning the GIS procedure for B-spline:

*[To avoid anomalies in the stacking of the facies, the interpolation is carried on their thickness instead of their boundaries' z-coordinates. In addition, the first underlying facies is not interpolated but considered as the background (filling) lithology.] The thicknesses of the four remaining facies are delivered in raster format, with integer values between 0 and 10 (i.e. the number of layers in the final 3D model), and a resolution of 200 m. Eventually, the facies stacking is transcribed for each column of prismatic elements in the 3D flow/transport mesh for TRACES according to the same majority analysis as for the GemPy procedure.*

TRACES and PINOGRI share the same code language and the same mesh structure (apart from the vertical dimension).

The figures caption require more complete descriptions.

More information has been provided in each figure's caption.

in 3,2: Differences in the facies composition of the models are marginal. ok but are they correct/reliable to be incorporated in the model?

The purpose of the flow/transport simulations is to verify whether the level of error produced by the interpolations (combined with the optimization error on the facies parameters) is compatible with the use of the method for practical purposes (reproduction of piezometric records and pollution plumes).

How the maximal discrepancy is 61 % of the initial value for the permeability and other increased discrepancies mentioned interprets a successful reconstruction of the aquifer dynamics. Obviously the model parameterization works well. Please explain clearly.

Compared to the previous version of the article, the optimization step has been changed in two ways:

The lithological constraints for the 3D optimization being only taken on location where the log is available, the potential interpolation errors are not reported anymore at this stage. Therefore, the optimization produces only one set of facies parameter per sampling (in this case, this step became independent from the lithology interpolation).

The optimization now integrates the 2D parameter uncertainty.

With this new setup, the maximal discrepancy (only considering the optimized value, and not its interval of confidence) is 45.5 % of the initial value, in a linear scale (namely 6.8x10-4 m/s vs 1.25x10-3x10 m/s). In addition, the confidence interval intersects the reference value (see Fig. 6 in Section 3.3). This kind of discrepancy is a small gap in the field of hydrogeology and at our scale of study. The illustrations with the piezometric series (and to a lesser extent the transport results) confirm it.

In figure 7 there are several dashed lines color, please elaborate. In addition, the aim of the work is a 3d hydrogeological characterization combining inversion, interpolation and optimization. It is not clear what fig 7 adds to the work and interpolation techniques application to water level. From the method and flowchart it seems that the interpolation applies to lithological data only.

To make the results figures more readable, now only the sparse sampling outputs are reported, with their confidence intervals (dotted and dashed envelope curves). These confidence intervals are produced by the extreme values amongst 4 supplementary simulations where the parameter values are set: (i) at the upper bound of the confidence interval, (ii) at the lower bound, (iii) alternately at the lower (for the facies 1,3,5) and upper (facies 2,4) bounds, (iv) alternately at the lower (for the facies 2,4) and upper (facies 1,3,5) bounds of the parameter confidence interval.

The water heads and contamination data are here to validate the methodology, i.e. to assess its ability to reproduce state variable of interest in the field of hydrogeology.

The level of consistency between the piezometric chronicles in particular attests that the discrepancies produced beforehand (during facies interpolation and parameter optimization) are marginal or acceptable at least.

line 400: Comparatively to joint inversion methods, the need of data acquisition and the computation efforts are lower. I do not fully agree, here you have uncertainty that you have not discussed. Stating this you need to support it with an example or literature.

The difference between our method and joint inversion in terms of data acquisition is stated in the introduction. There is no classical 3D approach that can rely on piezometric series only, without the help of direct measurement or geophysical surveys. Moreover, computation and even more inversion of 3D models is costlier in CPU than 2D models (with an otherwise equal level of refining on the horizontal dimensions).

reproduction of this work by the readers is not clear, please show some guidelines.

The first part of "Materials and methods" sums up the methodology. To reproduce it, one has to (i) estimate transmissivity from a 2D calibrated flow model, (ii) interpolate borehole data to obtain a 3D facies model, (iii) estimate the individual facies parameter through an optimization algorithm comparing the 2D transmissivity from (i) and the 3D transmissivity resulting from the optimized values.

The model results seems good in the analysis but a sensitivity analysis and uncertainty analysis in my personal opinion is needed to support the findings.

Uncertainties concerning the 2D inversion (i.e. the dispersion of transmissivities amongst the different solutions) and the optimization (i.e. 95 % interval of confidence for the facies parameters) are provided.

A more comprehensive uncertainty propagation and a sensitivity analysis will be performed on a following work.